# LUGS: Latent-aware Guidance for Efficient Unmasking in Diffusion Large Language Models

**Nuanqiao Shan** [1]  **Kairong Han** [1]  **Xinpeng Dong** [1]  **Kun Kuang** [1]

## Abstract

Diffusion Language Models (DLMs) have emerged as a flexible alternative to autoregressive (AR) models. They can decode tokens in any order, but the generation quality critically depends on the decoding strategy. Existing approaches predominantly rely on local heuristics, such as confidence or entropy, which may fail to capture sequence-level dependencies and the semantics in the context. To solve this problem, we propose Latent-aware Unmasking Guidance Search (LUGS), a novel decoding framework that leverages the model's internal hidden states to guide the unmasking process. By incorporating latent-aware scores to compensate for the limitations of local heuristics such as confidence or entropy, LUGS improves the model's performance. Extensive experiments on various downstream tasks demonstrate that our approach consistently outperforms existing baselines on LLaDA-8B-instruct and LLaDA-1.5 models. In Science and Reason tasks, LUGS improved performance by more than 1% on both base models. And LUGS obtains an average improvement of 3.5% in code generation. Remarkably, LUGS outperforms the beam search baseline by more than 5% on average using LLaDA-8B-Instruct on code tasks. These results highlight the potential of latent-aware guidance for advancing controllable and high-quality generation.

## 1. Introduction

The landscape of Large Language Models (LLMs) has been predominantly shaped by autoregressive (AR) models, which generate text by sequentially predicting tokens from left to right (Brown et al., 2020; Touvron et al., 2023; Achiam et al., 2023). To break the constraint of this fixed left-to-right decoding pattern, Diffusion Language Models (DLMs) have recently emerged as a promising alternative paradigm (Sahoo et al., 2024; Lou et al., 2023).

Recent large-scale DLMs such as LLaDA (Nie et al., 2025) and Dream (Ye et al., 2025) have demonstrated competitive performance on various downstream tasks, including text generation, code synthesis, and mathematical reasoning, which matches AR models of comparable size. DLMs' formulation naturally enables tokens to be revealed in any order during the iterative denoising process. However, a crucial challenge is how to select positions to unmask to achieve better performance. Although there are already some works that focus on the unmasking order, mostly based on local heuristics like confidence or entropy, they may fail to capture enough information to make the right choice. Choosing sub-optimal positions early in the process can lead to error accumulation, where the model commits to low-quality tokens that constrain the remaining generation into incoherent or repetitive patterns. There are already some works to address this problem. Some works (Sahoo et al., 2024; Shi et al., 2024; Nie et al., 2025) adopt greedy unmasking strategies. Some also employ methods based on beam search (Huang et al., 2025; Lee et al., 2025). These works mostly use logit-based information as scores to guide the unmasking process. But ignore the semantic information stored in the model's hidden states.

In this work, we propose **LUGS** to address the informationally impoverished nature of existing unmasking strategies. Our method improves the construction of candidate pools by integrating a Latent-aware Scoring Module ($f_\theta$). This module is a high-capacity Transformer-based architecture that leverages the last-layer hidden states of the DLMs. It takes in DLMs' last layer hidden states, temporal and mask embeddings, then gives out scores for every masked position to guide unmasking process.

An overview of the LUGS framework is illustrated in Figure 1. We divide the inference process into three steps. First, the DLM is used to perform a normal forward step. Then, the token logits and the hidden states from the last layer of the DLM are collected. We then use this information to

---

[1]College of Computer Science and Technology, Zhejiang University, Hangzhou, China. Correspondence to: Kun Kuang <kunkuang@zju.edu.cn>.

*Proceedings of the 43rd International Conference on Machine Learning*, Seoul, South Korea. PMLR 306, 2026. Copyright 2026 by the author(s).

compute local heuristic scores. Here, we insert the module to compute a guidance score to mitigate the limitation of local scores. Then, the final selection score is used to generate positions to be unmasked next.

However, it is challenging to train such a scorer due to a lack of groundtruth paths. We address this by employing a wide beam search on the training dataset to generate gold trajectories from training data. We then penalize the scorer if it fails to give gold candidate positions high scores. By emulating the selection logic, LUGS distills the decision strategy of the gold trajectories into a lightweight, high-capacity neural module.

During inference, the learned scorer($f_\theta$) guides the formation of the candidate pool, while efficient heuristics handle the pruning of sub-nodes. This synergy allows LUGS to achieve higher performance than the original Beam Search methods. Specifically, it improves the average accuracy in the Science and Reasoning tasks by 1.6% and the average Pass@1 in Code tasks by 3.54% across model variants.

Our main contributions are summarized as follows:

1. **Validation of Semantic Importance**: We empirically verify the critical role of semantic information in the unmasking decision process. Our analysis reveals that the high-level semantics encoded in hidden states exhibit a non-linear relationship with local heuristics. Latent-aware guidance provides unique, complementary signals that logit-based metrics fail to capture.

2. **A Latent-aware Decoding Framework**: We propose **LUGS**, a novel beam search framework for DLMs that transcends traditional heuristic-based unmasking. LUGS uses a high-capacity scoring module that interprets the model's last-layer hidden states to mitigate local heuristic limitations and identify strategically critical unmasking positions. We also introduce a training methodology that aligns a student scoring module with the decision strategy of the gold trajectories. Through a margin-based ranking objective at the step level, our module learns to prioritize positions that lead to high-quality global trajectories, distilling complex search knowledge into a lightweight guidance head.

3. **Better Performance on Various Tasks**: We conduct extensive evaluations across many challenging tasks, including mathematical reasoning (GSM8K, Math500), code generation (HumanEval, MBPP), logic puzzles (Countdown), and scientific commonsense (ARC-Challenge). LUGS shows consistent performance gains across all these domains, proving its robustness in handling various structured reasoning tasks.

**Conflict of Interest Disclosure**    The authors declare that there are no financial conflicts of interest for this work.

## 2. Related Work and Preliminaries

**Diffusion Language Models**    Diffusion models have achieved remarkable success in continuous domains such as image generation (Ho et al., 2020; Song et al., 2020), and have been increasingly adapted to discrete text sequences. Early attempts focused on continuous embeddings (Li et al., 2022) or discrete state transitions via categorical distributions (Austin et al., 2021a; Hoogeboom et al., 2021). Masked diffusion models represent a particularly successful variant, where generation proceeds by iteratively predicting and revealing masked tokens. MDLM (Sahoo et al., 2024) demonstrated that simple masked diffusion with proper training achieves competitive perplexity with AR models.

Recently, the focus has shifted toward large-scale masked language modeling as a form of discrete diffusion. Models such as LLaDA (Nie et al., 2025) and Dream (Ye et al., 2025) have demonstrated that by scaling parameters and training on massive corpora, DLMs can achieve performance parity with AR models. In the broader landscape of DLMs, extensive research has unfolded across multiple dimensions. From a theoretical and sampling perspective, studies have developed continuous-time frameworks (Lou et al., 2023; Campbell et al., 2022) and refined noise schedules (Austin et al., 2021a; Shi et al., 2024) to enhance sample quality and controllability. In the realm of multimodal generation, masked modeling has become a dominant paradigm for image and video synthesis (Chang et al., 2022), further inspiring unified any-to-any generation models (Xie et al., 2024; Hu et al., 2022). Additionally, hybrid approaches have been proposed to combine the speed of parallel decoding with the coherence of AR priors, often utilizing diffusion for iterative refinement (Savinov et al., 2021; Han et al., 2023). Furthermore, recent works integrate causal mechanisms and thought-masking strategies to enhance DLM reasoning and controllability (Han et al., 2025).

DLMs treat generation as an iterative refinement process, offering inherently parallelizable decoding and the potential for non-monotonic generation. However, the increased flexibility of DLMs introduces the complexity of determining the optimal unmasking order, a challenge that our work specifically addresses.

**Decoding Strategies**    The decoding process in DLMs is critical for generation performance. In the early stage, the research focus was directed towards training objectives and noise schedule designs. Most works adopt a simple random unmasking schedule (Hoogeboom et al., 2021), which provides a baseline but may be ineffective in complex reasoning or long-form generation. To improve on this, greedy strategies were introduced, leveraging token-level confidence or entropy to select the most certain positions at each step (Sahoo et al., 2024; Shi et al., 2024; Nie et al., 2025). Although

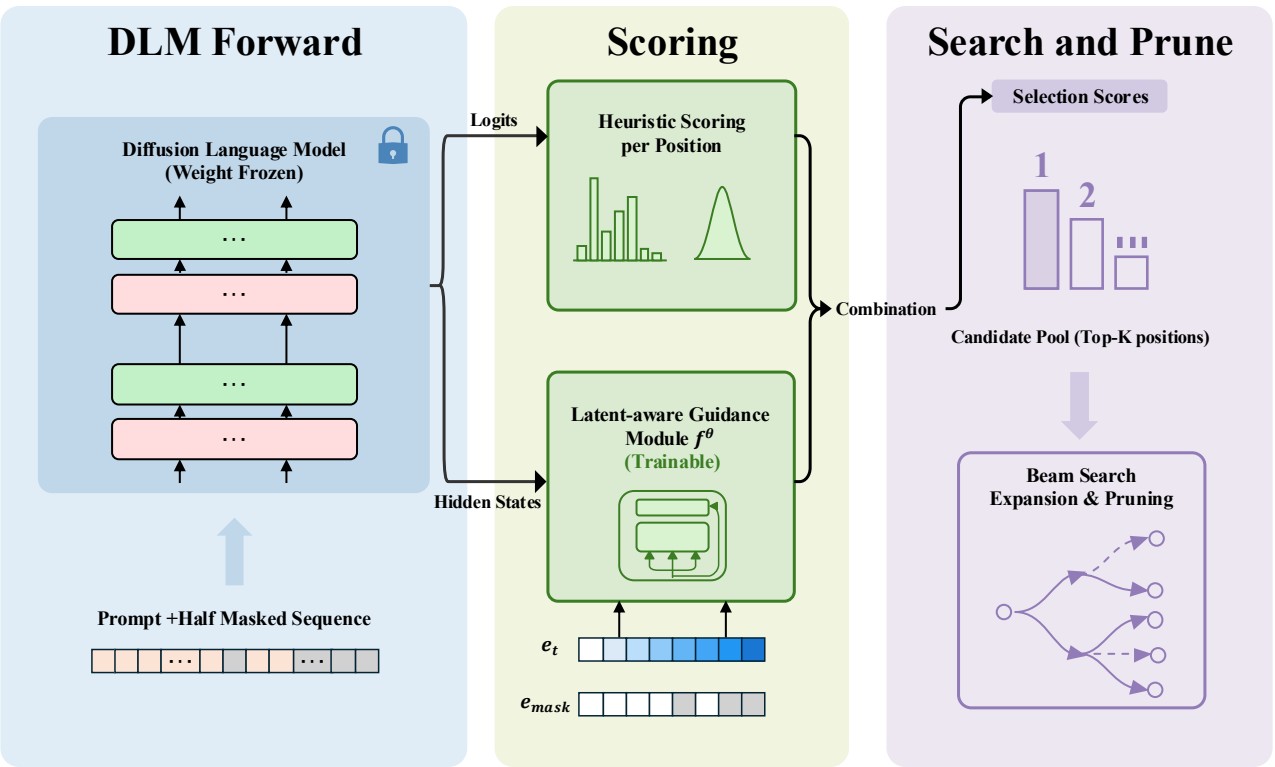

*Figure 1.* Overview of the proposed LUGS framework. At each decoding step, the frozen DLM produces token logits and last-layer hidden states. A latent-aware guidance module leverages the hidden states to score masked positions, which is combined with heuristic scores to form a candidate pool for beam search expansion. Here, $e_t$ serves as the timestep embedding.

these methods are efficient in most cases, they are myopic. That means early high-confidence but semantically inconsistent choices can lead to global degradation.

Then, some standard decoding methods move beyond these myopic strategies. They have introduced look-ahead mechanisms and beam search to explore multiple generation paths. For example, LookUM (Lee et al., 2025) introduced lookahead verification, using additional forward passes to evaluate unmasking decisions based on downstream uncertainty. Some frameworks (Huang et al., 2025) are also related. These methods use sequence-level uncertainty or simulated future steps to guide the search. However, they are bound to surface-level logit information.

There are also works that use deeper semantic information. For example, Jazbec et al. (2025) proposed learning unmasking policies via reinforcement learning, training a policy to decide how many tokens to unmask at each step. But this work is mainly about decoding acceleration.

To balance the trade-off between parallel efficiency and long-range coherence, semi-autoregressive (SAR) or block-wise decoding strategies have been adopted in recent DLMs. Following the LLaDA-style decoding framework (Nie et al., 2025), the total generation length $L_g$ is partitioned into

contiguous blocks of length $L_b$. The decoding proceeds sequentially block-by-block; however, within each block, the model performs $S$ denoising steps. Specifically, a per-step budget is set as $B = L_b/S$ (assuming $L_b$ is divisible by $S$). At each step, the model unmasks $B$ positions restricted to the current block. This approach defines a semi-autoregressive schedule (Han et al., 2023), while preserving non-monotonic flexibility.

**Search Guidance and Trajectory Optimization** Guiding the generation process with auxiliary modules is a well-established technique in reinforcement learning and AR decoding, typically implemented via value functions or reward models (Ouyang et al., 2022).

In the context of DLMs, early exploration of guidance primarily focused on classifier-based steering for controllable text generation (Li et al., 2022). However, optimizing the unmasking trajectory itself remains a relatively open challenge. Recent works have attempted to incorporate look-ahead mechanisms to refine generation paths (Lee et al., 2025).

LUGS diverges from these approaches by incorporating beam search for systematic exploration and extracting guid-

ance from the model's high-dimensional latent space, capturing semantic cues that logits alone might obscure.

## 3. Methodology

This section presents the LUGS decoding framework and its training recipe. We first define the masked diffusion decoding setup and the block-wise schedule, then describe the beam search inference procedure. We next introduce the latent-aware guidance module's architecture and the overall search algorithm. We finally illustrate the offline trajectory distillation objective used to train the module.

### 3.1. Notation Setup

We consider a discrete DLM that generates a sequence of length $L = L_p + L_g$, where the first $L_p$ tokens represent a prompt, and the remaining $L_g$ tokens are generated. Let $\mathcal{V}$ be the vocabulary and [MASK] be the special mask token in the masked positions. At any step $t$, the state of the sequence is $x^{(t)} \in (\mathcal{V} \cup \{[\text{MASK}]\})^L$. The DLM defines conditional token distributions $p_\phi(x_i \mid x^{(t)})$ for all masked positions $i$. The decoding process involves determining an unmasking trajectory over $T$ discrete steps. Starting from $x^{(0)}$, where all $L_g$ positions are masked, the model iteratively replaces [MASK] tokens with elements from $\mathcal{V}$ until a fully unmasked sequence $x^{(T)} \in \mathcal{V}^L$ is obtained at the final step $T$.

We also use semi-AR decoding. We partition the $L_g$ generation positions into blocks of length $L_b$ and perform $S$ denoising steps per block. We set the per-step budget as $B = L_b/S$, and unmask $B$ positions at each step.

### 3.2. Beam Search Decoding

We formulate decoding as a beam search where each node represents a partially unmasked sequence that has a fixed length of $L$. Figure 2 illustrates the beam search decoding process in LUGS. Starting from a fully masked sequence, we expand each active node by selecting $B$ masked positions from the pool of size $K$, and filling them with greedy token predictions. We expand each node into $R$ candidate children. This $R$ is also called branch factor. We keep a beam of width $W$. Then, to avoid maintaining too many children, we compute a node reward $d$ for every child on the sequence level and carry out the pruning process to keep $W$ candidates with the highest reward. We then initiate a new round of decoding. The details are described in Algorithm 1.

**Heuristic scores.** We first need to build a pool and select the unmasking positions in it afterwards. To build this pool,

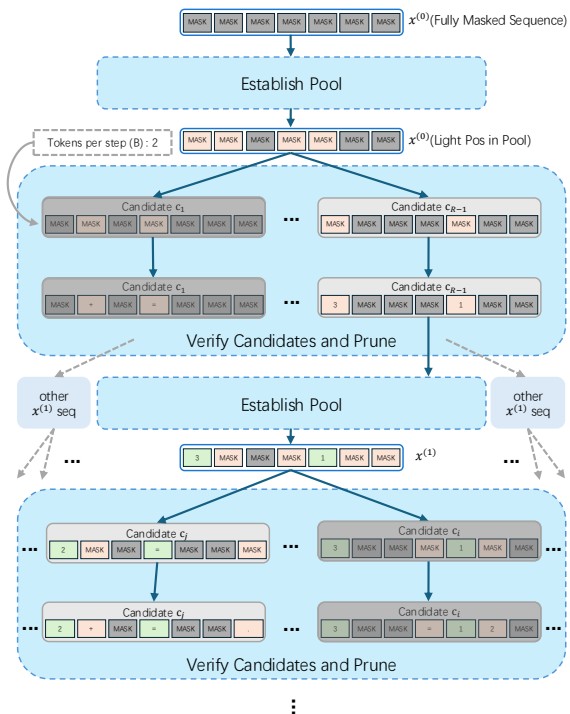

*Figure 2.* Beam search decoding procedure in LUGS. At each step, masked positions are ranked by latent-aware selection scores, a candidate pool is formed, and multiple child sequences are expanded and pruned based on sequence-level verification scores.

---

**Algorithm 1** LUGS Inference (Beam Search with Latent-aware Guidance)

---

**Require:** DLM $p_\phi$, guidance module $f_\theta$, prompt tokens $x_{1:L_p}$, generation length $L_g$, block length $L_b$, steps per block $S$, beam width $W$, branch factor $R$, budget $B = L_b/S$, pool size $K$

1: Initialize $x^{(0)}$ with prompt tokens and [MASK] elsewhere; set active beam $\mathcal{B} \leftarrow \{x^{(0)}\}$

2: **for** each block $b = 1, \ldots, L_g/L_b$ **do**

3:     **for** step $t = 1, \ldots, S$ **do**

4:         Expand each $x \in \mathcal{B}$:

5:             compute logits and hidden states

6:             compute heuristic $h_i$ and latent score $r_i = f_\theta(\cdot)$

7:             compute selection scores $s_i$ via Eq. (3)

8:             get top-$K$ selection scores and form pool $C_t$

9:             sample $R$ candidates by choosing $B$ positions from $C_t$ and filling with greedy tokens

10:            score each candidate with sequence-prediction based scores $d(\tilde{x})$ in Eq. (2)

11:        Update $\mathcal{B}$ by keeping top-$W$ candidates by sequence-prediction based score

12:    **end for**

13: **end for**

14: **return** best decoded sequence in $\mathcal{B}$

we compute a local heuristic score that combines confidence and entropy, with weights $w_p$ and $w_e$, respectively, for each masked position $i$ :

$$h_i = w_p \max_v p_\phi(x_i{=}v \mid x^{(t)}) + w_e \left(1 - \bar{H}_i\right), \quad (1)$$

where $\bar{H}_i = H_i / \log |\mathcal{V}|$ and $H_i = -\sum_v p_\phi(x_i{=}v \mid x^{(t)}) \log p_\phi(x_i{=}v \mid x^{(t)})$.

We use the scores to keep the top-$K$ positions in the pool. And we randomly select $B$ positions to create child nodes. Then these positions in the child nodes are unmasked.

**Sequence-prediction based verification.** We cannot let the child nodes proliferate uncontrollably because this will lead to excessive memory consumption. We have to prune the ones that are seen as not optimal. For each candidate child state $\tilde{x}$, we run one forward pass and compute a sequence-level heuristic score, where $j$ indexes all positions in the sequence:

$$d(\tilde{x}) = \frac{w_p}{L} \sum_{j=1}^{L} \max_v p_\phi(x_j = v \mid \tilde{x}) + \frac{w_e}{L} \sum_{j=1}^{L} \left(1 - \bar{H}_j(\tilde{x})\right) \quad (2)$$

It is important to be aware that scores from masked positions are also included in the computation of this heuristic score. We use $d(\tilde{x})$ as the node reward, keep $W$ candidates with the highest scores, and prune the others.

### 3.3. Latent-aware Guidance Module

We introduce a latent-aware scoring module $f_\theta$ that consumes the DLM's last-layer hidden states $H^{(t)} \in \mathbb{R}^{L \times d}$, a mask indicator $m \in \{0,1\}^L$, and a normalized timestep $\tau_t = t/T$. The module follows a LLaDA-style architecture: a sinusoidal timestep embedding, mask embedding, adaptive layer normalization, and a Transformer block, producing a score $r_i$ for each position during the pool formulation.

We combine latent-aware scores with the heuristic via a multiplicative gate:

$$s_i = h_i \cdot \left(\beta + (1 - \beta) \, \sigma(r_i)\right), \quad (3)$$

where $\sigma(\cdot)$ is the sigmoid function and $\beta \in [0, 1]$ controls the strength of the guidance. The pool $C_t$ is formed by selecting the top-$K$ masked positions ranked by $s_i$, and branches are generated by choosing $B$ positions from $C_t$.

### 3.4. Offline Trajectory Distillation

Training the guidance module requires supervision over unmasking choices, yet no ground-truth unmasking order is available. The trajectories are too long to use trivial reinforcement learning methods, because it is hard to get

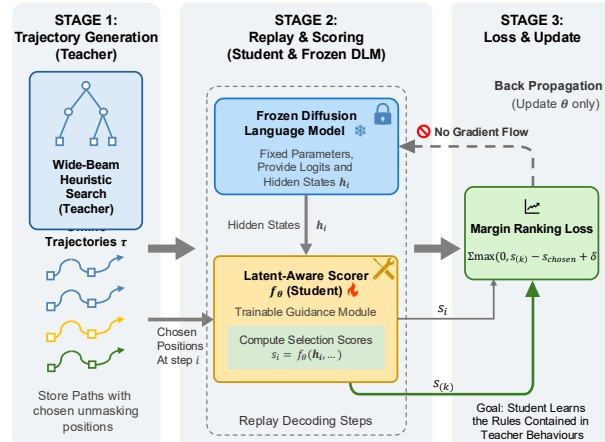

*Figure 3.* Offline trajectory distillation for training the latent-aware guidance module. A wide-beam heuristic decoder acts as a teacher to generate offline unmasking trajectories. During training, decoding states are replayed step by step to compute selection scores, and a margin-based ranking loss is applied to encourage the guidance module to recover the teacher's unmasking decisions. Only the guidance module is updated, while the underlying DLM remains frozen.

enough reward signals. To address this, we adopt an offline trajectory distillation strategy. As illustrated in Figure 3, a wide-beam heuristic decoder serves as a teacher to generate unmasking trajectories, from which a compact latent-aware guidance module is trained. Each trajectory records the chosen unmasking positions at every step.

Given a trajectory step with selected positions $\mathcal{P}_t$, we replay the decoding state to obtain $H^{(t)}$ and compute selection scores $\{s_i\}$. We then enforce that selected positions fall into the top-$K$ candidate positions via a margin-ranking loss:

$$\mathcal{L}_t = \sum_{i \in \mathcal{P}_t} \max(0, s_{(K)} - s_i + \delta), \quad (4)$$

where $s_{(k)}$ is the $k$-th largest selection score and $\delta$ is a margin. This step-level objective distills the search policy of a wide-beam teacher into a compact latent-aware module while keeping the underlying DLM frozen. In other words, only the guidance module is trained, which ensures the efficiency of the training process and maintains a strong architectural decoupling from the original DLM. The detailed training process is illustrated in Algorithm 2.

At implementation, we train on the full unmasking sequence of each trajectory instead of individual steps. By following the pre-recorded path from the initial fully-masked state to the final output, we can incrementally obtain the hidden states $H^{(t)}$ and supervise the guidance module at every step $t$ within one continuous pass. This approach eliminates the need to re-initialize the DLM for each training target, ensuring high computational throughput.

---

**Algorithm 2** LUGS Offline Trajectory Distillation

---

**Require:** DLM $p_\phi$ (frozen), guidance module $f_\theta$, offline trajectories $\{\tau\}$, pool size $K$, margin $\delta$
1: **for** each trajectory $\tau$ **do**
2:     Initialize $x^{(0)}$ as fully masked with prompt; load step records $\{\mathcal{P}_t\}$
3:     **for** step $t$ in $\tau$ **do**
4:         Replay state to obtain $x^{(t)}$ and hidden states $H^{(t)}$
5:         Compute $h_i, r_i = f_\theta(\cdot)$, and selection scores $s_i$
6:         Compute loss $\mathcal{L}_t$ in Eq. (4) using top-$K$ threshold
7:         Update $\theta$ with $\nabla_\theta \mathcal{L}_t$
8:     **end for**
9: **end for**

---

## 4. Experiments

To comprehensively assess the robustness of LUGS, we conduct experiments in a diverse set of structured reasoning tasks, ranging from mathematical problem solving and code generation to logical deduction and scientific common sense. Our primary objective is to assess whether the proposed latent-aware guidance provides improvements over standard heuristic-based decoding strategies. In the following sections, we detail our experimental setup, present comparative results against varied baselines, and provide an in-depth analysis of the interaction between latent guidance and inference dynamics.

### 4.1. Experimental Setup

**Base Models and Baselines** We implement our framework on two representative DLMs: **LLaDA-8B-Instruct** and **LLaDA-1.5** (Zhu et al., 2025). These models serve as the backbone for calculating logits and extracting hidden states for our guidance module.

We evaluate LUGS against a comprehensive set of decoding strategies, ranging from greedy heuristics to unguided search methods. For greedy approaches, we include **Random** unmasking as the most basic method, alongside metric-based strategies that prioritize positions based on **Probability** (highest token confidence), **Entropy** (lowest predictive uncertainty), and **Margin** (the largest gap between top-1 and top-2 probabilities). Crucially, to isolate the contribution of our learned guidance module, we compare against a **Standard Beam Search without Guidance**. This baseline utilizes the same beam width and search structure as LUGS but constructs candidate pools relying solely on standard heuristics like Probability or Entropy rather than our latent-aware scoring function.

**Tasks and Metrics.** We evaluate performance across four distinct domains to ensure robustness. Table 1 shows the datasets and relative information. Specifically,

*Table 1.* Datasets and evaluation metrics. We report **Pass@1** for code generation tasks and **Accuracy** for all other reasoning tasks.

| DATASET | DOMAIN | METRIC |
|---|---|---|
| GSM8K | MATH | ACCURACY |
| MATH500 | MATH | ACCURACY |
| ARC-EASY | SCIENCE | ACCURACY |
| ARC-CHALLENGE | SCIENCE | ACCURACY |
| HUMANEVAL | CODE | PASS@1 |
| MBPP | CODE | PASS@1 |
| COUNTDOWN | LOGIC | ACCURACY |

we select tasks spanning mathematics, code generation, logic, and scientific commonsense. For mathematical reasoning, we employ **GSM8K**(Cobbe et al., 2021) and **Math500**(Lightman et al., 2023), a representative subset of the **MATH** (Hendrycks et al., 2021), and use exact answer matching to calculate the accuracy. Specifically, for the MATH task, we train our guidance module on the full training split and evaluate on the Math500 subset. In the code domain, we evaluate on **MBPP**(Austin et al., 2021b) and **HumanEval**(Chen et al., 2021) using the Pass@1 metric. Notably, our guidance module is trained solely on the MBPP training set and evaluated on both datasets to test cross-distribution generalization. We also include the **Countdown** task to assess logical planning and **ARC-Easy/Challenge** (Clark et al., 2018) for scientific common sense, and compute the accuracy based on the validity of the equation and the selection of options, respectively.

**Implementation Details.** We follow the block-wise decoding schedule. Specifically, we set the block length $L_b = 32$ and perform $S = 16$ steps per block, unmasking $B = 2$ tokens per step.

For search parameters, we use a Beam Width $W = 3$, a Branching Factor $R = 3$, a Pool Size $K = 5$, and Guidance Weight $\beta = 0.5$.

The guidance module is trained on subsets of the respective task training data. We generate ground-truth training trajectories using a wider beam search. To be exact, we set $W = 10$, $R = 10$, Margin $\delta = 0.1$, to capture high-quality unmasking paths in this case. We use 30000 trajectories from the corresponding tasks for training. Specifically, for HumanEval and Math 500, which do not have their own train split, we use MBPP and Math for training, respectively. The generation length $L_g$ is 512 for code tasks, in this case, HumanEval and MBPP. And $L_g$ is 256 for other tasks. We set $L_g$ in this way because code tasks may need longer sequences to accommodate the inherent verbosity of programming syntax and ensure the functional completeness of the generated solutions. In this work, all experiments were run on $8\times$ NVIDIA H100 GPUs.

## 4.2. Main Results

Table 2 summarizes the performance of LUGS compared to the baselines across all tasks. Overall, LUGS outperforms the baseline methods on both base models in most cases, demonstrating the effectiveness of latent-aware guidance in enhancing decoding quality. Notably, in the experiments based on the LLaDA-8B-Instruct base model, LUGS outperforms unguided Beam Search by 7.31% on the humaneval dataset, and achieves an average improvement of 5.05% across all code-related tasks. On LLaDA-1.5, it also shows an enhancement of 2.02% in code-related tasks. This further demonstrates that LUGS holds greater potential in code generation. There are also improvements in the Science and Reasoning tasks on both base models.

## 4.3. Effect of Beam Width

In this section, we analyze the impact of beam width $W$ on both performance and inference latency. We vary $W$ from 2 to 9 and compare LUGS against the baseline on the Countdown, HumanEval, and MBPP tasks using LLaDA-8B-Instruct. The results are illustrated in Figure 4.

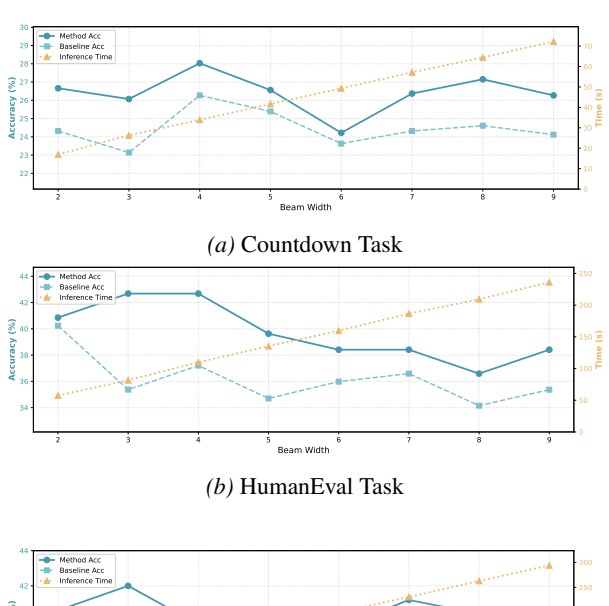

*(a)* Countdown Task

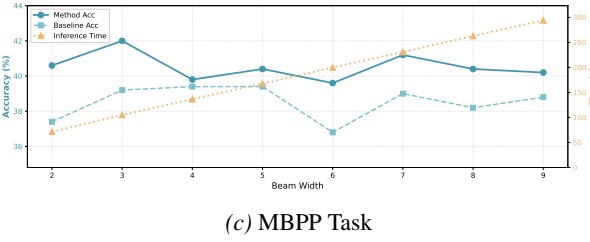

*(b)* HumanEval Task

*(c)* MBPP Task

*Figure 4.* **Accuracy vs. Inference Time across different Beam Widths.** The left axis (blue) shows the task performance (Accuracy/Pass@1) for LUGS (solid) and the Baseline (dashed). The right axis (orange) shows the inference time in seconds. LUGS consistently outperforms the baseline.

As shown in Figure 4, although the performance gains do not scale monotonically with beam width, LUGS consistently surpasses the baseline method across all tested beam widths. The detailed data is shown in Appendix A.2. Notably, in tasks of Countdown, HumanEval and MBPP, our method has an average improvement of 1.94%, 3.51%, and 2%, demonstrating that the latent-aware guidance effectively steers the generation towards correct trajectories much more efficiently than simply increasing the search space.

Although LUGS has better performance in all these cases, the time cost should also be considered. The inference time per data point increases almost linearly with the beam width, reflecting the computational cost of expanding more candidates. Based on these observations, we adopt $W = 3$ as the default setting in our main experiments to strike an optimal balance between performance and computational efficiency.

## 4.4. Impact of Guidance Weight

In Eq. (3), the hyperparameter $\beta \in [0, 1]$ controls the trade-off between the base heuristic score $h_i$ and the learned latent-aware guidance $\sigma(r_i)$. A smaller $\beta$ emphasizes the contribution of the latent module, while a larger $\beta$ relies more heavily on the original heuristic. To investigate the sensitivity of LUGS to this hyperparameter, we evaluated the performance on the Countdown task using the LLaDA-8B-Instruct backbone, varying $\beta$ from 0.0 to 1.0.

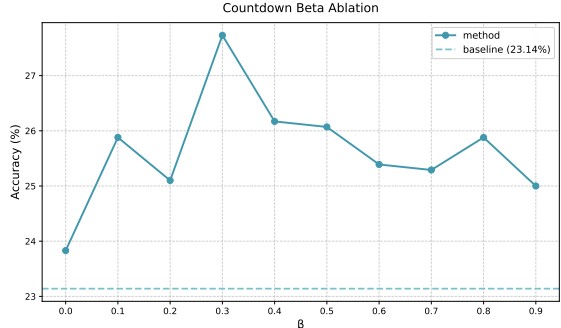

*Figure 5.* Sensitivity analysis of the guidance weight $\beta$ on the Countdown task. The dashed line represents the baseline Beam Search performance (23.14%) without latent guidance. LUGS consistently outperforms the baseline across a wide range of $\beta$.

Figure 5 illustrates the accuracy trends. We observe that introducing latent-aware guidance consistently improves performance over the standard Beam Search baseline (23.14%) across all tested values of $\beta$. At $\beta = 0.0$, the selection score is purely modulated by the latent output—the model achieves 23.83%. It still slightly surpasses the baseline.

The results suggest that the latent guidance module produces a crucial complementary signal to the local heuristic, and the method is robust to the hyperparameter $\beta$, especially within

*Table 2.* Main results comparison. We report Accuracy (%) for Math, Logic, and Commonsense tasks, and Pass@1 (%) for Code tasks. All Beam Search methods (including LUGS) use a beam width of $W = 3$. The best performance is highlighted in **bold**.

| Model | Method | Science & Reasoning | | | | | Code | | |
|---|---|---|---|---|---|---|---|---|---|
| | | GSM8K | Math500 | Countdown | ARC-C | Avg. | HumanEval | MBPP | Avg. |
| **LLaDA-8B -Instruct** | Random | 67.48 | 25.80 | 12.50 | 78.67 | 46.11 | 15.24 | 24.40 | 19.82 |
| | Probability | 78.32 | 33.20 | 19.34 | 82.34 | 53.30 | 37.20 | 35.60 | 36.40 |
| | Entropy | 77.10 | 32.00 | 14.65 | 81.74 | 51.37 | 31.71 | 33.60 | 32.66 |
| | Margin | 75.74 | 33.40 | 20.90 | **83.87** | 53.48 | 32.32 | 34.00 | 33.16 |
| | Beam Search | **79.38** | 33.60 | 23.14 | 83.02 | 54.79 | 35.37 | 39.20 | 37.29 |
| | LUGS (Ours) | **79.38** | **34.80** | **26.07** | 83.45 | **55.93** | **42.68** | **42.00** | **42.34** |
| **LLaDA-1.5** | Random | 68.54 | 25.40 | 11.43 | 81.57 | 46.74 | 11.59 | 30.00 | 20.80 |
| | Probability | 78.85 | 32.20 | 25.39 | 83.96 | 55.10 | 35.98 | 34.40 | 35.19 |
| | Entropy | 79.83 | 29.80 | 19.82 | 82.94 | 53.10 | 34.15 | 32.00 | 33.08 |
| | Margin | 78.32 | 33.20 | 23.73 | 84.47 | 54.93 | 32.32 | 32.80 | 32.56 |
| | Beam Search | 80.14 | 35.40 | 29.20 | 84.04 | 57.20 | 39.63 | 40.40 | 40.02 |
| | LUGS (Ours) | **81.27** | **37.60** | **30.08** | **84.56** | **58.38** | **42.68** | **41.40** | **42.04** |

the $[0.3, 0.8]$ interval. This empirical analysis validates the setting of $\beta = 0.5$ used in our main experiments (§4.2), confirming that it is within the optimal region for balancing heuristic scores and latent-aware guidance.

### 4.5. Capturing Richer Context

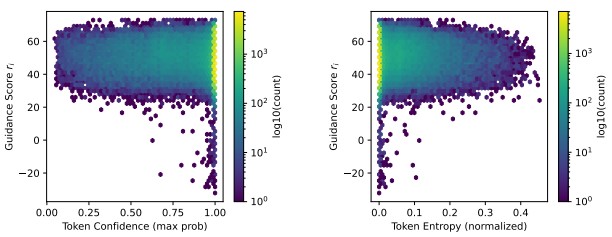

*(a)* Guidance vs. Confidence    *(b)* Guidance vs. Entropy

*Figure 6.* **Distribution of learned guidance scores.** The guidance score $r_i$ shows high variance when conditioned on **(a)** model confidence or **(b)** predictive entropy. This non-linear relationship demonstrates that LUGS captures semantic cues orthogonal to simple prediction certainty.

To verify that LUGS learns a search policy distinct from standard statistical heuristics, we analyze the relationship between the guidance scores $r_i$ and two standard metrics (confidence and entropy) on the token level. We record the tokens selected by LUGS during the decoding process in the Math500 dataset and visualize these relationships. Figure 6 display the joint distributions of guidance scores against confidence and entropy, respectively. We observe two clouds rather than a near-linear diagonal.

Figure 6(a) shows that the relationship between guidance scores and confidence is highly non-linear with significant variance. Notably, we observe cases with high confidence but low guidance scores. This implies that even if the model is certain about a token, the guidance module may suppress

it due to the semantic information it observes.

Similarly, the guidance score is not a simple inverse of entropy in Figure 6(b). The broad distribution in the upper-right region suggests that LUGS can identify positions that are uncertain but must be resolved early to anchor the subsequent generation. In contrast, some low-entropy tokens receive lower scores. This indicates that they can be safely deferred to facilitate global planning.

These deviations confirm that LUGS exhibits a high-variance, non-linear relationship with these metrics. $f_\theta$ is not distilling the fixed DLM distribution, but is learning a distinct search policy.

## 5. Conclusion

In this work, we introduced LUGS, a novel decoding framework designed to overcome the limitation of existing local heuristic methods in DLMs. By looking beyond surface-level logits and leveraging the high-dimensional semantic information within the model's hidden states, our latent-aware guidance module effectively identifies strategically critical unmasking positions that traditional heuristics sometimes overlook, and achieves better performance in various tasks. Through the offline trajectory distillation process, we successfully trained the guidance module to learn the unmasking strategies of a wide-beam search, thereby distilling search-derived insights into a lightweight and efficient scoring mechanism.

Our extensive empirical evaluation across mathematical reasoning, code generation, and logic tasks demonstrates that LUGS consistently outperforms existing greedy and heuristic-based beam search methods. Our guidance module is not merely recovering token confidence, but is capturing distinct, look-ahead semantic cues essential for structured generation.

## Acknowledgments

This work was supported in part by the National Key Research and Development Program of China (2024YFE0203700), National Natural Science Foundation of China (No. 62376243), the Key Research and Development Program of Zhejiang Province (2026C01021), and the "Pioneer" and "Leading Goose" R&D Program of Zhejiang (2025C02037). All opinions in this paper are those of the authors and do not necessarily reflect the views of the funding agencies.

## Impact Statement

This paper introduces LUGS, a framework that enhances the decoding efficiency and generation quality of DLMs. By focusing on algorithmic improvements for complex reasoning tasks like mathematics and code synthesis, this work provides a technical contribution to the field of non-autoregressive generation. This research does not introduce any ethical risks or adverse societal impacts.

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

# A. Additional Experimental Results

## A.1. Hyperparameter Settings in detail

*Table 3.* Hyperparameters for trajectory generation

| Group | Parameter | Value |
| --- | --- | --- |
| Data | Max samples | 3000 |
| Model | Base model | LLaDA-8B-Instruct & LLaDA-1.5 |
| Model | Precision | bfloat16 |
| Model | Mask token id | 126336 |
| Generation | Gen length $L_g$ | 256 for science & reasoning; 512 for code |
| Generation | Block length $L_b$ | 32 |
| Generation | Steps per block $S$ | 16 |
| Generation | Tokens per step $B$ | 2 |
| Generation | Total steps | 128 for science & reasoning; 256 for code |
| Search | Beam width $W$ | 10 |
| Search | Pool size $K$ | 5 |
| Search | branch factor $R$ | 10 |
| Heuristic | $w_p$ | 0.5 |
| Heuristic | $w_e$ | 0.5 |
| Decoding | Token choice | Argmax Greedy |
| Parallelism | GPUs | 8 |

*Table 4.* Hyperparameters for Guidance Module Training

| Group | Parameter | Value |
| --- | --- | --- |
| Data | Only correct | True |
| Data | Max trajectories | 3000 |
| Model | Base model | LLaDA-8B-Instruct & LLaDA-1.5 |
| Model | Precision | bfloat16 |
| Model | Mask token id | 126336 |
| Model | Gen length $L_g$ | 256 for science & reasoning; 512 for code |
| Model | Total steps | 128 for science & reasoning; 256 for code |
| Guidance Module | Hidden dim | 4096 |
| Guidance Module | Num heads | 32 |
| Guidance Module | FFN size | 14336 |
| Guidance Module | Num layers | 1 |
| Guidance Module | Timestep emb dim | 256 |
| Guidance Module | Dropout | 0.0 |
| Guidance Module | Block embedding | False |
| Training | Epochs | 2 |
| Training | Steps/trajectory | 128 for science & reasoning; 256 for code |
| Training | Validation samples | 80 |
| Training | Optimizer | AdamW |
| Training | Learning rate | 5e-6 |
| Training | Weight decay | 0.0 |
| Training | LR schedule | Constant |
| Training | Distributed | DDP (NCCL) |
| Loss | Loss type | Top-5 margin |
| Loss | Margin $\delta$ | 0.1 |
| Heuristic | $w_p$ | 0.5 |
| Heuristic | $w_e$ | 0.5 |

*Table 5.* Hyperparameters for Main Experiment Evaluation

| Group | Parameter | Value |
|---|---|---|
| Data | Max samples | full |
| Model | Base model | LLaDA-8B-Instruct & LLaDA-1.5 |
| Model | Precision | bfloat16 |
| Model | Mask token id | 126336 |
| Generation | Gen length $L_g$ | 256 for science & reasoning; 512 for code |
| Generation | Block length $L_b$ | 32 |
| Generation | Steps per block $S$ | 16 |
| Generation | Tokens per step $B$ | 2 |
| Generation | Total steps | 128 for science & reasoning; 256 for code |
| Search | Beam width $W$ | 3 |
| Search | Branch factor $R$ | 3 |
| Search | Pool size $K$ | 5 |
| Heuristic | $w_p$ | 0.5 |
| Heuristic | $w_e$ | 0.5 |
| Guidance Score | $\beta$ | 0.5 |
| Guidance Score | Use Hidden states | True |
| Guidance Score | Keep hidden on GPU | True |
| Guidance Module | Hidden dim | 4096 |
| Guidance Module | Num heads | 32 |
| Guidance Module | FFN size | 14336 |
| Guidance Module | Num layers | 1 |
| Guidance Module | Timestep emb dim | 256 |
| Guidance Module | Block embedding | False |

## A.2. Additional information about Effect of Beam Width Experiment

*Table 6.* Performance comparison on Countdown, HumanEval, and MBPP datasets. The best results are highlighted in **bold**.

| Countdown | | | HumanEval | | | MBPP | | |
|---|---|---|---|---|---|---|---|---|
| Time (s) | Base (%) | Method (%) | Time (s) | Base (%) | Method (%) | Time (s) | Base (%) | Method (%) |
| 16.96 | 24.32 | **26.66** | 57.52 | 40.24 | **40.85** | 71.50 | 37.40 | **40.60** |
| 26.28 | 23.14 | **26.07** | 81.45 | 35.37 | **42.68** | 104.97 | 39.20 | **42.00** |
| 33.91 | 26.27 | **28.03** | 109.89 | 37.20 | **42.68** | 136.63 | 39.40 | **39.80** |
| 41.63 | 25.39 | **26.56** | 135.22 | 34.70 | **39.63** | 167.62 | 39.40 | **40.40** |
| 49.31 | 23.63 | **24.22** | 159.70 | 35.98 | **38.41** | 199.93 | 36.80 | **39.60** |
| 57.14 | 24.32 | **26.37** | 186.49 | 36.59 | **38.41** | 230.85 | 39.00 | **41.20** |
| 64.45 | 24.61 | **27.15** | 209.26 | 34.15 | **36.59** | 262.90 | 38.20 | **40.40** |
| 72.27 | 24.12 | **26.27** | 236.02 | 35.37 | **38.41** | 294.04 | 38.80 | **40.20** |
| **Avg.** | 24.48 | **26.42** | - | 36.20 | **39.71** | - | 38.53 | **40.53** |

## A.3. Additional information about Impact of Guidance Weight Experiment

*Table 7.* Accuracy(%) about Impact of Guidance Weight Experiment on Countdown

| $\beta$ | **Accuracy (%)** |
|---------|-------------------|
| 0.0 | 23.83 |
| 0.1 | 25.88 |
| 0.2 | 25.10 |
| 0.3 | 27.73 |
| 0.4 | 26.17 |
| 0.5 | 26.07 |
| 0.6 | 25.39 |
| 0.7 | 25.29 |
| 0.8 | 25.88 |
| 0.9 | 25.00 |
| **Baseline** | **23.14** |

## B. Detailed Illustration Picture of Beam Searching

In the main body, we do not have enough space to place this full picture. Instead, we use Figure 2 there. Figure 7 is the full version.

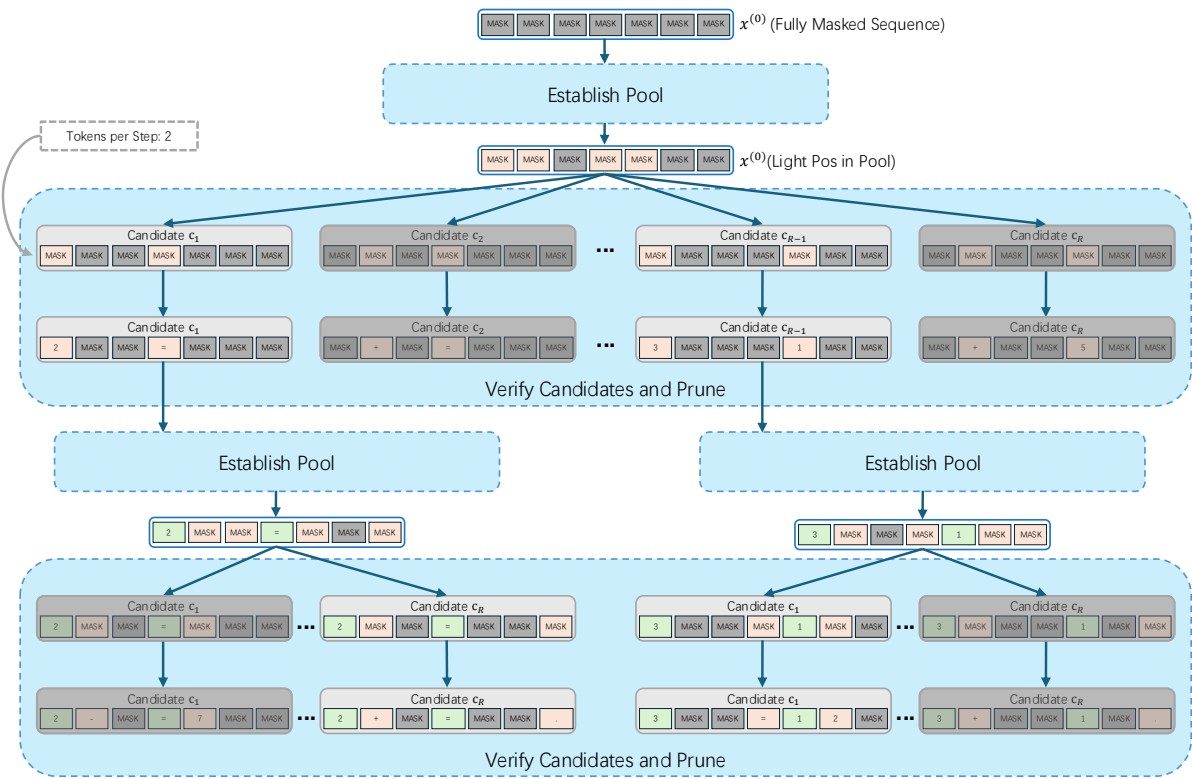

*Figure 7.* The full picture of beam search in DLMs

## C. Computational Cost

The peak memory usage during training is approximately 22.39 GiB per GPU. During inference, we observe that LUGS incurs only a small overhead compared to the beam search baseline. Specifically, the peak memory usage of LUGS is 18.11 GiB per GPU, compared to 16.77 GiB for beam search ($\approx 1.08\times$). In terms of runtime, LUGS is approximately $1.07\times$ the time of beam search under the same setting. The runtime of the beam search baseline is reported in Figure 4, making this comparison directly grounded in our experimental results.

## D. Cross-Task Generalization Analysis

To evaluate how well the learned guidance module generalizes to entirely unseen tasks, we conducted cross-task experiments under the same setting as the main experiments in the paper. First, we consider cross-category generalization, where the training and testing tasks belong to different task types (e.g., from MBPP to Countdown). The results are summarized in Table 8. We further examine within-category generalization, where the model is transferred between tasks of the same type. For example, we evaluate transferring from GSM8K to Math500. The results are shown in Table 9.

*Table 8.* Cross-category generalization results. Training and testing tasks belong to different task types.

| Training Dataset | Test Dataset | Score | In-domain (trained on target task) |
| --- | --- | --- | --- |
| MBPP | Countdown | 28.12 | 26.07 |
| MBPP | Math500 | 36.00 | 34.80 |
| Countdown | HumanEval | 45.12 | 42.68 |
| Countdown | MBPP | 40.60 | 42.00 |

*Table 9.* Within-category generalization results. Transfer between tasks of the same type.

| Training Dataset | Test Dataset | Score | In-domain (trained on target task) |
| --- | --- | --- | --- |
| GSM8K | Math500 | 35.20 | 34.80 |
| GSM8K | Countdown | 27.15 | 26.07 |
| Math | Countdown | 28.03 | 26.07 |
| Countdown | Math500 | 35.00 | 34.80 |

These results suggest that the learned guidance module maintains reasonable performance or even achieves higher performance in most cases when applied to unseen tasks, including both cross-category and within-category transfers. Although in the case we transfer from Countdown to MBPP, it shows a slight performance gap compared to the in-domain result, it still outperforms the beam search baseline in Table 2.

## E. Future work

Looking ahead, several promising directions emerge. While LUGS improves the sample quality of DLMs, integrating it with acceleration techniques like speculative decoding could further reduce inference latency. There is also space to investigate reinforcement learning techniques to directly optimize the guidance module against sequence-level rewards—rather than distilling from trajectories. Finally, incorporating a gate mechanism into local heuristics and guidance may also be a promising direction. We believe LUGS represents a significant step toward making non-autoregressive generation robust and reliable for complex reasoning tasks.

