# OpenReview forum: "LUGS: Latent-aware Guidance for Efficient Unmasking in Diffusion Large Language Models"
_ICML.cc/2026/Conference — ICML 2026 regular_

### Official Review · Reviewer_yFv3 · 2026-03-05

**Soundness:** 3
**Presentation:** 3
**Significance:** 2
**Originality:** 3
**Overall Recommendation:** 4
**Confidence:** 3

**Summary:**

The paper proposes a new method to determine which tokens should be predicted next in diffusion LLM. The authors assume that the position that has low entropy and high prediction probability for a token should be predicted first (see (1)). The authors conduct a beam search according to this heuristic score and distill the results of the beam search to a smaller guidance module f^theta in Figure 1. The results show that combining the scores predicted by the guidance module and the heuristic scores is significantly better than only using the heuristic scores in the beam search.

**Compliance With Llm Reviewing Policy:**

Affirmed.

**Final Justification:**

Since all my comments are addressed well, I would like to raise my final score from 4 to 4.5. I agree with other reviewers. The improvement is not very large, so the significance might not be very large. But everything else, such as soundness, is good enough for being accepted.

**Key Questions For Authors:**

1. Your training reward basically comes from the heuristic scores from (2) for the beam search. The heuristic score could be computed at each step during the testing time. Then, why do you think distilling such a signal to a smaller network is helpful? Does the improvement come from better optimizing (2) using the beam search, or come from some smoothing effect of your smaller network? I think the paper could have some more experiments and/or discussion on this question.

2. What is e_t in Figure 1? I cannot find e_t in section 3.3

**Limitations:**

Yes

**Strengths And Weaknesses:**

Soundness Strength:

The method is relatively simple. The experiment setups are reasonable, and improvements are significant. Overall, the results and analyses make sense and justify the proposed method. Nevertheless, I still list some questions below to confirm the soundness of the work.

---------------
Significance Strength

It looks like a promising approach, which suggests that learning a guidance module for LLaDA could improve the performance. This opens some new research opportunities to study how to make this guidance module better in the future.

---------------

Significance Weakness:

The rewards of the guidance module basically come from the heuristic scores d(x~) using (2), which limit its potential performance gains. It would be better if the rewards could come from verifiable functions (i.e., 270-273-left should be treated as future work).

Furthermore, I have two main concerns for this paper: generalization and inference speed. 314-right said "The guidance module is trained on subsets of the respective task training data." and the authors do not provide the results of OOD setup (i.e., training on one task/dataset and testing on another/dataset). Furthermore, the authors do not compare the inference speed of their method and baseline in Figure 4, so we don't know how much cost we need to pay to run that guidance module. These two issues are intertwined. To achieve good generalization ability, I think training a larger module on many tasks/datasets is unavoidable, which will slow down the inference speed.


---------------

Originality Strength:

Although predicting the generation order for language models is not a new idea, as the related work section shows, I did not see a method that distills the beam search based on heuristic scores. The paper proposes a new method for LLMs that works well, which is increasingly difficult nowadays. I am not an expert on diffusion LM, so I am not sure if there are papers proposing similar methods. If not, the novelty of this paper is sufficient in my opinion.

---------------

Presentation Strength:

Overall presentation is clear and easy to understand. I list some minor suggestions below.



Minor presentation suggestions:

Figure 1: The next move should just be one step of beam search, but the flowchart seems to suggest that it is doing the whole beam search to determine the next move

034-right beam search( -> beam search (

252 tau_t refers to normalized timestep but tau is also used to represent the trajectory

---

> ### Author Rebuttal · Authors · 2026-03-29
>
> > Q1
>
> We would like to clarify that our method is not simply distilling the heuristic score in Eq. (2), but rather learning a trajectory-level decision policy that cannot be directly recovered from test-time heuristics. First, our guidance module operates at the pre-expansion stage, deciding which positions to unmask before branching. This is fundamentally different from Eq. (2), which is computed at the post-expansion stage. We note that while trajectories are generated using beam search affected by Eq. (2), our supervision signal is not the heuristic score itself, but the selection of successful trajectories obtained after the full search process. Therefore, the model learns from which decisions survive the search process, rather than directly approximating Eq. (2). Second, our empirical results support this interpretation. If the improvement came from better optimizing Eq. (2), then standard beam search and LUGS should achieve similar performance. However, we observe gains in LUGS. Furthermore, Fig. 6 shows that the learned guidance scores have a highly non-linear and high-variance relationship with heuristic metrics, ruling out a simple smoothing effect.
>
> We further note that if our model merely approximated the heuristic score in Eq. (2), one would expect it to be tightly coupled to the specific task and distribution. However, our additional cross-task generalization experiments below for Q2 show that the learned guidance module transfers well across tasks, often matching or even exceeding in-domain performance. This suggests that the model captures a more generalizable search policy, rather than overfitting to a particular heuristic signal.
>
> In summary, our method does not approximate the heuristic score, but instead learns where to allocate the limited search budget, effectively distilling long-horizon search behavior into a lightweight policy.
>
> We will clarify this distinction and strengthen the discussion in the revision.
>
>
> > Q2
>
> e_t denotes the timestep embedding, which corresponds to the normalized timestep tau_t described in Section 3.3. We acknowledge that this notation was not clearly defined in the text, and we will revise the paper to make this explicit.
>
> > W1
>
> Thank you for pointing this out. We intend to explore more verifiable algorithms and leave this as future work.
>
>
>
> > W2
>
> To evaluate how well the learned guidance module generalizes to entirely unseen tasks and see the performance on the OOD setup, we conducted cross-task experiments under the same setting as the main experiments in the paper.
>
> First, we consider cross-category generalization, where the training and testing tasks belong to different task types (e.g., from MBPP to Countdown). The results are shown below.
>
> | Training Dataset | Test Dataset | Score | In-domain (trained on target task) |
> |------------|------------|------|-------|
> | MBPP       | Countdown       |    28.12  | 26.07 |
> | MBPP      | Math500      |   36.00   | 34.80 |
> | Countdown     | HumanEval      |   45.12   | 42.68 |
> | Countdown     | MBPP      |   40.60   | 42.00 |
>
> We further examine within-category generalization, where the model is transferred between tasks of the same type. For example, we evaluate transferring from GSM8K to Math500:
>
> | Training Dataset | Test Dataset | Score | In-domain (trained on target task) |
> |------------|------------|------|--------|
> | GSM8K       | Math500     |   35.20   | 34.80 |
> | GSM8K       | Countdown       |   27.15   |  26.07 |
> | Math       | Countdown       |   28.03   |  26.07 |
> | Countdown        | Math500      |   35.00   |  34.80 |
>
> These results suggest that the learned guidance module maintains reasonable performance or even achieves higher performance in most cases when applied to unseen tasks, including both cross-category and within-category transfers. Although in the case we transfer from Countdown to MBPP, it showed a slight performance gap compared to the in-domain result, it still outperforms the beam search baseline in Table 2. This confirms that the guidance module remains highly effective even in the most challenging transfer scenarios. We will include these comprehensive generalization results in the revised manuscript.
>
>
> > W3
>
> To clarify the computational cost, we have measured the inference speed of our method compared to the baselines. LUGS is approximately 1.07× the time of beam search under the same setting. We will include this in the final version of the manuscript.
>
> > Suggestions
>
> Thank you for these valuable suggestions. We have carefully addressed all the minor presentation issues. All corresponding changes have been made in the revised manuscript. We appreciate your attention to detail.

---

> > ### Author Rebuttal · Reviewer_yFv3 · 2026-04-02
> >
> > Thanks for the rebuttal. It is impressive to see that the cross-domain performance could sometimes be better. Since all my comments are addressed well, I would like to raise my final score from 4 to 4.5.

---

> > > ### Author Response · Authors · 2026-04-07
> > >
> > > We sincerely appreciate your positive feedback and your decision to raise the score.

---

### Official Review · Reviewer_rSCc · 2026-03-10

**Soundness:** 2
**Presentation:** 3
**Significance:** 2
**Originality:** 3
**Overall Recommendation:** 3
**Confidence:** 3

**Summary:**

This paper proposes LUGS, a decoding framework for masked diffusion language models that improves the choice of which masked token positions to reveal at each step. It adds a learned latent-aware scoring module that uses the frozen model’s last-layer hidden states (plus timestep/mask signals) to guide candidate position selection within a beam search, and trains this module by distilling decisions from offline trajectories generated by a wider heuristic search. Experiments on LLaDA models across reasoning and code benchmarks show consistent improvements over standard heuristic-based decoding and unguided beam search.

**Compliance With Llm Reviewing Policy:**

Affirmed.

**Final Justification:**

I summarize this work as using an additional bidirectional attention layer to guide the unmasking process, which shows a modest yet consistent improvement in the accuracy of dLLMs

**Key Questions For Authors:**

1. What new information does the latent-aware scoring head provide beyond logits?
The DLM’s logits are already computed from the last-layer hidden states, and “local heuristics” (confidence/entropy) are functions of these logits. It is therefore unclear why adding an extra scoring head over the same hidden states fundamentally overcomes the limitation of “local heuristics,” rather than just learning another proxy for logit-based uncertainty.
2. How do you rule out that gains mainly come from additional task-specific training/distillation rather than “hidden-state guidance”?
Because the scoring head is trained on task data via offline trajectory distillation, improvements on coding tasks could be partially (or mostly) due to this extra supervised adaptation, not necessarily due to the proposed latent-aware unmasking insight.
3. What evidence supports generalization of a scoring head trained on relatively small data?
The guidance module is trained on a limited number of trajectories compared to typical LLM training scales. How robust is it to domain shift, different prompts/lengths, or different tasks?

**Limitations:**

yes

**Strengths And Weaknesses:**

Strengths
1. The paper introduces a relatively novel idea: using the DLM’s hidden states (latent semantic information) to guide unmasking decisions, and implements a complete pipeline around it (trajectory generation, training a guidance scorer, and evaluation on multiple tasks).
2. The writing is generally clear and the overall logic is easy to follow; the problem motivation and the proposed framework are explained in a coherent way.

Weaknesses
1. On non-coding tasks, the gains (1%) over the baseline are relatively small, which may suggest the hidden-state guidance is not strongly or consistently correlated with better unmasking decisions across domains (at least under the current setup).
2. The figures are visually rough and sometimes not very intuitive, which makes it harder to quickly understand the decoding/search procedure.
3. The method requires training an additional scoring head (and generating teacher trajectories), which increases compute/engineering cost and raises the barrier to adoption compared with purely heuristic decoding.

---

> ### Author Rebuttal · Authors · 2026-03-29
>
> > Q1
>
> Logit-based methods, such as entropy or confidence, are restricted to fixed, token-wise functions of the output distribution. But our scoring head operates directly on the hidden representation, which encodes rich contextual and cross-token information. This allows it to learn a general function that is not constrained to any predefined statistic of the logits, enabling the model to capture decision-relevant signals.
>
> We also conduct an experiment in which we train two scoring heads with identical architectures and training procedures: one operating on logits, and the other on hidden states. Despite both showing better performance compared to the beam baseline, the logits-based model slightly underperforms lugs.
>
> |  | ours | logits |
> |------|-------|-------|
> | Countdown | 26.07 | 25.00 |
> | math500 | 34.80 | 34.20 |
> | mbpp | 42.00 | 39.40 |
> | humaneval | 42.68 | 39.02 |
>
>
>
> > Q2
>
> | Training Dataset | Test Dataset | Score | In-domain (trained on target task) |
> |------------|------------|------|-------|
> | MBPP       | Countdown       |    28.12  | 26.07 |
> | MBPP      | Math500      |   36.00   | 34.80 |
> | Countdown     | HumanEval      |   45.12   | 42.68 |
> | Countdown     | MBPP      |   40.60   | 42.00 |
>
> These results suggest that the learned guidance module maintains reasonable performance or even achieves higher performance in most cases when applied to unseen tasks in a cross-category scenario. Although in the case we transfer from Countdown to MBPP, it showed a slight performance gap compared to the in-domain result, it still outperforms the beam search baseline in Table 2. This result suggests that the learned guidance does not rely on task-specific patterns or memorization, but instead captures a more generalizable unmasking strategy applicable across different generation problems. If the gains were primarily due to task-specific distillation, we would expect them to diminish under such cross-domain transfer, which is not observed.
>
>
> > Q3
>
> The guidance module learns a decision policy over latent representations from a DLM. The module captures reusable search strategies rather than memorizing dataset-specific patterns.
>
> Empirically, our main results already demonstrate strong robustness across domains(e.g., MBPP to HumanEval), where LUGS improves over baselines despite distribution shifts.
>
> To further directly evaluate generalization, we additionally show the results where we train the guidance module on one task and evaluate it on different unseen tasks.
>
> | Training Dataset | Test Dataset | Score | In-domain (trained on target task) |
> |------------|------------|------|--------|
> | GSM8K       | Math500     |   35.20   | 34.80 |
> | GSM8K       | Countdown       |   27.15   |  26.07 |
> | Math       | Countdown       |   28.03   |  26.07 |
> | Countdown        | Math500      |   35.00   |  34.80 |
>
> The cross-task model results showed even better performance compared to the In-domain results. This provides strong evidence that the learned guidance is not overfitting to a specific dataset, but instead captures transferable principles for effective unmasking. We will include these additional results and clarifications shown in response to Q2 & Q3 in the revision.
>
> > W1
>
> We would like to emphasize that our method shows improvements across most of the tasks, including both mathematical reasoning and code generation. We also note that the Countdown task, which is not a code task and requires strong numerical reasoning, exhibits relatively larger gains, further supporting the effectiveness of our method in structured reasoning settings. We believe the gains on different tasks reflect task-dependent sensitivity to unmasking strategies.
>
> More importantly, the primary contribution of this work is to propose a new decoding paradigm. We agree that further engineering improvements may amplify gains on non-coding tasks. However, the improvements across most of the tasks have validated the effectiveness and generality of this design.
>
>
>
> > W2
>
> Thank you for this helpful suggestion. We have revised and improved the figures in the paper to enhance their clarity and visual quality.
>
> > W3
>
> The training overhead of LUGS is modest. The peak memory usage during training is approximately 22.39 GiB per GPU. During inference, we observe that LUGS incurs only a small overhead compared to the beam search baseline. Specifically, the peak memory usage of LUGS is 18.11 GiB per GPU, compared to 16.77 GiB for beam search (≈1.08×). In terms of runtime, LUGS is approximately 1.07× the time of beam search under the same setting. The runtime of the beam search baseline is reported in Figure 4, making this comparison directly grounded in our experimental results. Despite involving multiple candidate expansions and sequence-level verification, the actual computational overhead of LUGS is limited and does not lead to a significantly higher cost in practice. The additional overhead of LUGS remains modest and practically acceptable.

---

> > ### Author Rebuttal · Reviewer_rSCc · 2026-04-01
> >
> > I summarize this work as using an additional bidirectional attention layer to guide the unmasking process, which shows a modest yet consistent improvement in the accuracy of dLLMs

---

> > > ### Author Response · Authors · 2026-04-01
> > >
> > > Thank you very much for your time and for the acknowledgement.

---

### Official Review · Reviewer_DeYQ · 2026-03-11

**Soundness:** 2
**Presentation:** 3
**Significance:** 3
**Originality:** 3
**Overall Recommendation:** 4
**Confidence:** 4

**Summary:**

The paper proposes LUGS, a novel decoding framework for diffusion language models. To overcome the limitations of local heuristics in determining the unmasking order, LUGS introduces a lightweight scoring module that leverages the model's last-layer hidden states. Experiments demonstrate consistent performance improvements over standard heuristic baselines.

**Compliance With Llm Reviewing Policy:**

Affirmed.

**Final Justification:**

The rebuttal addresses most of my concerns. I will keep my score leaning toward accept.

**Key Questions For Authors:**

1. Can you provide a more detailed breakdown of the computational overhead (e.g., time, peak memory usage) of LUGS compared to standard greedy decoding and other accelerated samplers?
2. How does the performance and efficiency of LUGS compare against recent advanced sampling techniques?
3. How well does the learned guidance module generalize to entirely unseen tasks?

**Limitations:**

Yes.

**Strengths And Weaknesses:**

**Strengths:**
1. The integration of beam search with a trained latent-aware scoring function to guide the unmasking process is an interesting approach.
2. The proposed method demonstrates performance gains across diverse domains.

**Weaknesses:**
1. The method seems to have a significantly larger computational cost. The inference time scales almost linearly with the beam width due to the expansion of multiple candidates and sequence-level verifications. It is unfair to directly compare with other baselines.
2. The evaluation primarily compares LUGS against basic heuristics and standard beam search. It lacks comparisons with recent state-of-the-art DLM samplers, such as EB-Sampler (Ben-Hamu et al., 2025).
3. There are a few minor errors that require proofreading (e.g., missing "." in line 420).

---

> ### Author Rebuttal · Authors · 2026-03-29
>
> > Q1 & W1
>
> The training overhead of LUGS is modest. The peak memory usage during training is approximately 22.39 GiB per GPU. During inference, we observe that LUGS incurs only a small overhead compared to the beam search baseline. Specifically, the peak memory usage of LUGS is 18.11 GiB per GPU, compared to 16.77 GiB for beam search (≈1.08×). In terms of runtime, LUGS is approximately 1.07× the time of beam search under the same setting. The runtime of the beam search baseline is reported in Figure 4, making this comparison directly grounded in our experimental results. Despite involving multiple candidate expansions and sequence-level verification, the actual computational overhead of LUGS is limited and does not lead to a significantly higher cost in practice. The additional overhead of LUGS remains modest and practically acceptable. We will include this information in the revised manuscript.
>
> > Q2 & W2
>
> Thank you for pointing out the importance of comparing with recent advanced samplers such as EB-Sampler(Ben-Hamu et al., 2025). However, we would like to clarify that LUGS is not designed as an acceleration method. Instead, it focuses on improving search quality under a fixed decoding budget. Compared to EB-Sampler, which dynamically adjusts the number of tokens unmasked per step to reduce the number of decoding iterations, the number of tokens unmasked per step is fixed (e.g., 2 tokens) in LUGS. LUGS emphasizes better sequence-level search. As such, the two approaches differ in their design goals and operating regimes, making direct comparison under identical settings non-trivial. We agree that the adaptive unmasking strategy in EB-Sampler is an interesting direction and may inspire future extensions of our method. We have added a discussion of this work in the revised paper.
>
> While EB-Sampler excels at maintaining baseline performance with fewer NFEs, LUGS is designed to push the performance ceiling of DLMs. EB-Sampler maintains performance close to the baseline. In contrast, LUGS is designed to improve generation quality under a fixed decoding budget. In our experimental setting, we observe improvements over the baseline across most of the datasets.
>
> > Q3
>
> To evaluate how well the learned guidance module generalizes to entirely unseen tasks, we conducted cross-task experiments under the same setting as the main experiments in the paper.
>
> First, we consider cross-category generalization, where the training and testing tasks belong to different task types (e.g., from MBPP to Countdown). The results are shown below.
>
> | Training Dataset | Test Dataset | Score | In-domain (trained on target task) |
> |------------|------------|------|-------|
> | MBPP       | Countdown       |    28.12  | 26.07 |
> | MBPP      | Math500      |   36.00   | 34.80 |
> | Countdown     | HumanEval      |   45.12   | 42.68 |
> | Countdown     | MBPP      |   40.60   | 42.00 |
>
> We further examine within-category generalization, where the model is transferred between tasks of the same type. For example, we evaluate transferring from GSM8K to Math500:
>
> | Training Dataset | Test Dataset | Score | In-domain (trained on target task) |
> |------------|------------|------|--------|
> | GSM8K       | Math500     |   35.20   | 34.80 |
> | GSM8K       | Countdown       |   27.15   |  26.07 |
> | Math       | Countdown       |   28.03   |  26.07 |
> | Countdown        | Math500      |   35.00   |  34.80 |
>
> These results suggest that the learned guidance module maintains reasonable performance or even achieves higher performance in most cases when applied to unseen tasks, including both cross-category and within-category transfers. Although in the case we transfer from Countdown to MBPP, it shows a slight performance gap compared to the in-domain result, it still outperforms the beam search baseline in Table 2. This confirms that the guidance module remains highly effective even in the most challenging transfer scenarios. We will include these comprehensive generalization results in the revised manuscript.
>
> > W3
>
> Thank you very much for pointing this out! We have carefully checked the details in the paper and made the necessary corrections.

---

> > ### Author Rebuttal · Reviewer_DeYQ · 2026-04-03
> >
> > Thanks for your rebuttal. However, regarding computational overhead, I mean in comparison to the other baselines in Table 2, such as Probability and Margin. I think it is somewhat unfair to make a direct comparison, since your method uses roughly three times more compute. What if these baselines were also run three times with different hyperparameters (e.g., temperature, block size), and the best result were selected?

---

> > > ### Author Response · Authors · 2026-04-07
> > >
> > > To address this concern, we conducted additional experiments where we increased the computational overhead of the baseline methods to match that of LUGS. Specifically, we adjusted the hyperparameters for these baselines—including block_length, steps_per_block, and tokens_per_step—such that their inference time is now aligned with LUGS (with a marginal difference of less than 5%).
> > > The results under this equal-compute setting are summarized in the table below:
> > > | Method |	Math500 | Countdown |
> > > |----|----|----|
> > > | Entropy |	34.20 | 3.22 |
> > > | Probability | 34.80 | 11.13  |
> > > | Random | 25.80 | 7.03  |
> > > | LUGS (Ours) | 34.80  | 26.07  |
> > >
> > > LUGS maintains a substantial advantage in more complex reasoning scenarios such as Countdown. We believe these additional experiments directly address your concern and demonstrate that LUGS provides a fundamental algorithmic improvement over standard baselines beyond simple compute scaling.

---

### Decision · Program_Chairs · 2026-04-30

**Decision:**

Accept (regular)

**Comment:**

LUGS proposes a learned latent-aware scoring module to guide token unmasking order in diffusion LMs, showing consistent if modest gains over heuristic baselines with good cross-task generalization. Scores are 4/3/4; Reviewer rSCc's concerns were fully resolved and maintained weak reject but acknowledged modest consistent improvement. The main remaining concern — that compute-matched probability sampling matches LUGS on Math500 — is valid and should be addressed in the revision, but the overall panel is positive.